# A Comparative Study of Different Poly (Lactic Acid) Bio-Composites Produced by Mechanical Alloying and Casting for Tribological Applications

**DOI:** 10.3390/ma16041608

**Published:** 2023-02-15

**Authors:** Anzum Al Abir, Bruno Trindade

**Affiliations:** CEMMPRE—Centre for Mechanical Engineering, Materials and Processes, University of Coimbra, Rua Luís Reis Santos, 3030-788 Coimbra, Portugal

**Keywords:** PLA-based bio-composites, mechanical alloying, casting, hardness, friction, wear

## Abstract

The aim of this study was to fabricate different self-lubricating poly (lactic acid)-based bio-composites reinforced with mono- and multi-fillers of carbon fibers, graphene nanoparticles, and a soft Sn-based brazing alloy (Sn89-Zn8-Bi3) using a two-step process consisting of mechanical alloying followed by casting. The results showed that the incorporation of the different fillers on the PLA surface by mechanical alloying was quite homogenous. The volume ratio between the PLA and the fillers was 1:0.02, respectively. The PLA sample reinforced with short carbon fibers and graphene nanoparticles presented the highest hardness (84.5 Shore D, corresponding to a 10% increase compared to PLA) and the lowest specific wear rate (1.5 × 10^−4^ mm^3^/N·m, one order of magnitude lower than PLA). With regard to the coefficient of friction, the lowest value was obtained for the sample reinforced with graphene (0.43, corresponding to a decrease of 12% compared to PLA).

## 1. Introduction

Polymer-based materials have been gradually replacing other materials in many engineering applications, such as construction, aerospace, automotive, and biomedical [1]. However, in most of these applications, petroleum-based polymers have been used, with harmful effects on the environment. In the last few years, there has been a tendency to replace them with biodegradable polymers. However, in general, biodegradable polymers present lower mechanical and tribological properties that impede their use in most structural applications. By incorporating appropriate compatible reinforcements, researchers hope to create biodegradable polymer composites that can satisfy a variety of practical needs [2,3]. The poly (lactic acid) (PLA) polymer has a significant potential in a large range of applications in different fields of engineering, such as in mechanical parts (gears), biomedical engineering, and electronic engineering [4,5,6]. PLA is a thermoplastic biodegradable polymer, and its degradation products are also non-toxic to humans and the environment. Moreover, the production of PLA uses much less energy than the petroleum-based polymers [7]. However, similar to most biopolymers, PLA has some limitations, due to its low mechanical properties, low thermal resistance, and high wear rate [8]. The combination of PLA with nanoparticles, such as carbon-based, cellulose nanocrystals (CNCs) and metallic fillers, is one of the possible ways to produce high-performance PLA bio-nanocomposites [9]. The influence of carbon mono-fillers, such as carbon nanotubes (CNT) [10,11], multi-walled carbon nanotubes (MWCNT) [12], short carbon fibers (CSF) [13], and graphene nanoplatelets (GNP) [14,15,16,17] on polymer matrixes has been studied by different researchers. Mixtures of different carbon forms have also been used as multi-fillers to reinforce PLA. Examples of this are: multi-wall carbon nanotubes (MWCNT) and GNP [12], continuous carbon fibers (CCF) and short carbon fibers (SCF) [13], and carbon fibers and graphene oxide (GO) [18]. In most studies, it is stated that, in general, the addition of carbon-based mono- or multi-fillers increase the mechanical and tribological properties of PLA. Batakliev [12] investigated the influence of MWCNTs and GNP on the tribological behavior of reinforced PLA filaments produced by melt extrusion. The results showed that the PLA exhibited a coefficient of friction (COF) in the range of 0.15–0.16, whereas the composite sample with 3 wt.% of GNP presented the lowest COF (0.07). Maqsood et al. [13] studied the mechanical properties of PLA and PLA reinforced with CCF and SCF. The results showed a significant increase in the tensile strength and the Young’s modulus for the composite samples. In a recent study of 3D-printed graphene/carbon fiber multi-scale-reinforced PLA composites [18], the authors showed that the mechanical properties of such composites prepared by the mechanical stripping method were significantly improved by adding graphene oxide to the mixture.

Despite there being several studies available on carbon nanoparticle-reinforced PLA composites, there has been no significant investigation of the incorporation of both GNP and SCF into a PLA matrix and their effects on the mechanical (hardness) and tribological (friction and wear) properties. It is expected that the joint addition of these fillers to PLA may increase its performance, since it is known that graphene is a solid lubricant, leading to better tribological properties (lower COF and wear); the SCF increases both properties. 

Literature about the effect of metallic particles on a PLA matrix is scarce. Fafenrot et al. [19] studied the influence of bronze and magnetic iron on the mechanical properties of PLA. More recently, Hamidi et al. [20] and Vakharia et al. [21] added metal fibers (copper, bronze, and steel wire) into PLA. Liu et al. [22] assessed the mechanical response of Cu- and Al-reinforced PLA. In all of these works, enhanced strength of the composites was reported. 

The present study concerns the production of PLA samples reinforced with SCF, GNP, and an Sn89-Zn8-Bi3 alloy. This alloy was chosen because of its biocompatibility with the human body [23,24]. A novel and simple technique consisting of mechanical alloying (MA) followed by casting was used to produce these composite samples. MA was used to synthesize the metallic alloy from metallic elemental powders and to coat the PLA granules with the reinforcements. The main objective of using this solid-state processing technique was to achieve a homogeneous distribution of the fillers around the PLA granules. Conventional casting was used to fabricate the reinforced PLA composites. The samples were characterized by different experimental techniques, and the results are presented and discussed.

## 2. Materials and Methods

Table 1 shows the samples produced in this research. The composite samples were fabricated in two stages: mechanical alloying (MA) of the PLA granules with GNP, SCF, and an Sn-based alloy, followed by casting. The PLA granules (ρ = 1.25 g/cm^3^) were supplied by Goodfellow. The Sn_89_-Zn_8_-Bi_3_ alloy was fabricated by MA from elemental powders of Sn (purity 99.9%), Zn (purity 98.8%), and Bi (purity 99.5%) supplied by Goodfellow. The SCF (ρ = 1.8 g/cm^3^) and the GNP (density of 2 g/cm^3^) fillers were provided by Sigrafil and Nanografi, respectively.

The MA process was carried out in a planetary ball mill (Frisch Pulverisette 6) to produce the composites prior to casting. The rotation speed was 300 rpm. The ball-to-powder ratio was 20:1. A 500 mL capacity Cr-tempered steel bowl and 15 grinding balls with a diameter of 10 mm of the same material were used. The synthesis of the Sn89-Zn8-Bi3 alloy was performed with a maximum milling time of 25 h. The bowl was opened after 5, 10, and 25 h to collect some powder for morphological and structural characterization. The PLA + fillers were mechanically alloyed for 10 min to produce the composite samples. The relative amount of the SCF and GNP fillers was 50–50 (wt.%). Additionally, 1 wt.% stearic acid was added as a process control agent (PCA) to avoid excessive cold welding of the metallic powders.

For the casting process, a 4 cm diameter mold was used. The amount of material poured into the mold was 4.6 g. The casting process was performed at 200 °C, with a dwell time of 1 h. The heating rate was 5 °C/min. The samples were cooled down inside the furnace to a temperature of 35 °C to maintain the crystallinity of the PLA.

The samples were analyzed using different techniques. A scanning electron microscope (SEM) (Hitachi-SU3800) was used for the morphological analysis. An energy-dispersive X-ray spectroscope (EDS) (Oxford Instruments X-MaxN) was used for the chemical analysis. The structure of the raw particles and materials produced was examined by X-ray diffraction (XRD) (Philips XPert) with Co-Ka radiation (1.790300 Å). X-ray microtomography (SKYSCAN) was used to obtain 3D images of the cross-section of the coated PLA granules and to determine the thickness of the coatings produced by MA. Differential scanning calorimetry (DSC) (NETZSCH-DSC 204 F1 Phoenix) was used to study the thermophysical behavior of the PLA matrix from 25 to 230 °C. A heating rate of 10 °C /min and a N_2_ flux of 40 mL/min was used to perform these tests. The degree of crystallinity of the PLA was determined using the following equation [25]:*Xc* (%) = (Δ*Hf*/Δ*Hfo*) × 100(1)
where Δ*Hf* represents the melting enthalpy of the samples, and Δ*Hfo* is the standard enthalpy of PLA (93 J/g [25]).

Fourier-transform infrared spectroscopy (FTIR) (BRUKER—ALPHA2) was used to assess the PLA’s chemical properties before and after casting. The hardness of the cast samples was assessed by Shore D hardness (CV Instruments Limited, Sheffield, United Kingdom), using the ASTM D2240-00 standard. Five tests were performed on each sample.

The tribological properties of the samples were determined by the reciprocating ball on disk tests (Rtec instrument-MFT 5000). Before testing, the samples were polished using silicon carbide abrasive papers with successive grades up to 4000 mesh. A load of 5N was applied through an SS (100 Cr6) ball with a diameter of 5 mm. The stroke length was 6 mm, the frequency was 8.5 Hz, and the maximum time of each tribological test was 300 s. The experiments were performed at 25 °C, with a humidity of 50%. The wear tracks and scars on the samples were analyzed by 3D profilometry (Alicona-InfiniteFocus, Raaba/Graz, Austria).

The specific wear rate, k, was calculated using the equation [26]:*k* = *V*/(*N*·*S*) (m^3^/(N·m))(2)
where *V* is the wear volume on the sample, *N* is the load applied, and *S* is the distance of the journey.

## 3. Results

### 3.1. Raw Materials

Figure 1 shows the optical and SEM images of the raw materials.

The PLA granules (Figure 1a) had a more or less spherical shape, with an equivalent diameter of approximately 4.8 mm. The GNP powder (Figure 1b) had a flake-like shape, with stacked layers of graphite sheets; the SCF consisted of micrometer cylindrical rods, which were tens of micrometers in length and had a diameter of less than 10 μm (Figure 1c). The Bi particles (Figure 1d) had a spherical shape and identical sizes. The Zn particles (Figure 1e), despite being more or less spherical, were less regular than the Bi particles, and their size distribution was larger. Finally, the Sn particles (Figure 1f) presented a large size distribution, with a droplet-shaped spheroidal morphology. 

Figure 2 shows the XRD patterns of the raw materials used in this work.

The PLA had two intense diffraction peaks at 2θ = 19° and 22°, ascribed to the (200/100) and (203) planes, and two smaller peaks, corresponding to the (010) and (015) diffraction planes of the α′-form orthorhombic crystallographic structure [27]. The lattice parameters were calculated from the (010), (200), and (203) planes, and the values of a = 1.06 nm, b = 0.57 nm, and c = 2.85 nm were obtained. The Sn89-Zn8-Bi3 alloy showed diffraction peaks ascribed to the Sn phase. A Bi diffraction peak was visible at the 42.4° angle. The diffraction peaks placed at the angles of 45.7° and 50° corresponded to the Zn phase.

The DSC curve of the raw PLA is shown in Figure 3. It exhibited two peaks placed at 60 and 173 °C, corresponding to the glass temperate (Tg) and the melting temperature (Tm), respectively. After DSC analysis, the crystallinity (Xc) of the PLA was calculated using equation 1, and the value obtained was 45%.

### 3.2. Mechanical Alloying of the Sn89-Zn8-Bi3 Mixture

The SEM images of the Sn89-Zn8-Bi3 mixture after 5, 10, and 25 h of MA are presented in Figure 4.

During the first 5 h of MA, the particles plastically deformed and became flattened (Figure 4a) as a result of the high ductility of the raw metallic powders [28]. From 5 to 10 h of MA (Figure 4b), a decrease in the average particle size of the mixture occurred, resulting from a fracture process, due to the work hardening of the particles by plastic deformation [29]. After 25 h of MA (Figure 4c), the particles were quite irregular and of different sizes. It was concluded from the EDS elemental analysis that the metallic mixture MA’ed for 25 h was quite homogeneous chemically (Figure 5). Intense peaks of tin were present in the 25 h mechanically alloyed mixture, together with lower intensity peaks of Zn and Bi (lower concentrations in the mixture).

The XRD patterns of the Sn89-Zn8-Bi3 mixture as a function of the MA time are presented in Figure 6. After 5 h of MA, the diffraction peaks of the Sn, Zn, and Bi were still detectable. After 10 h of MA, the diffraction peaks of Zn were no longer detected, meaning that the dissolution of the Bi atoms into the Sn lattice had occurred. After 25 h of MA, the XRD pattern of the mixture showed only the diffraction peaks of Sn. It was likely that the Zn and Bi atoms were incorporated into the Sn lattice. However, no significant shift of diffraction peaks of Sn was observed, due to the similar atomic radii of these elements (1.45, 1.42, and 1.43 Å for Sn, Zn, and Bi, respectively). Similar results were obtained in a previous work on this system [30].

### 3.3. Mechanical Alloying of the PLA and Fillers

Figure 7a shows an optical micrograph of the PLA granules coated with GNP and Sn89-Zn8-Bi3 by MA as a typical example of all the other coated particles.

The composites were dark gray in color, meaning that there was a good coverage of the PLA surface by MA. Figure 7b,c shows the SEM images of a PLA granule mechanically alloyed with GNP and SCF. The corresponding EDS elemental map of the surface is shown in Figure 7d. The results confirmed that the incorporation of the different fillers on the PLA granules was successfully achieved, with quite a homogenous distribution. An average thickness of 20 μm was determined for the coatings by X-ray microtomography, corresponding to a volume ratio between the PLA and the fillers of 1:0.02, respectively.

### 3.4. Casting

All but the cast PLA samples had the same gray tone as the granules coated by MA. From the XRD analysis of the non-reinforced and reinforced PLA samples (Figure 8a), it was concluded that any structural transformations occurred during casting. However, the diffraction peaks of the cast PLA sample were slightly broader, and a large and low intensity peak centered at 19.5°, ascribed to a decrease in its structural order (partial amorphization), could be detected. The crystallinity (Xc) of the cast PLA sample was calculated, and a value of 40% was obtained, which was 5% lower than that for raw PLA. This confirms the PLA’s slight loss of crystallinity induced by the cooling process of the casting process [31]. The DSC analysis of the cast PLA corroborated the XRD results (Figure 8b). A cold crystallization peak at ~115 °C was detected above the glass transition temperature (Tg), which was not observed for the raw PLA. 

The FTIR results of the raw and cast PLA samples are shown in Figure 9. The IR spectra were quite similar, and only small variations of the intensity of the peaks were observed. This means that during casting, the polymer remained quite stable, and no changes in its chemical composition occurred. The IR spectra showed a strong band at 1750 cm^−1^, ascribed to the C=O bond stretching, some bands in the range of 3000 to 2850 cm^−1^, assigned to the C-H stretching of -CH_3_, and the characteristic absorption of ester C-O, which stretches at 1179 cm^−1^.

The XRD patterns of the cast samples are shown in Figure 10. The characteristic diffraction peak of GNP was placed at 2θ = 33°. The diffraction peak of the SCF (2θ = 31.8°) was superimposed on the PLA diffraction plane. Concerning the PA sample, the characteristic peaks of Sn could be seen in the diffraction pattern. Therefore, based on these results, one might conclude that all the fillers were present in the cast samples.

As mentioned before, hardness and tribological tests were performed on all the samples. The Shore D hardness values are presented in Figure 11. The hardness measured for PLA was 77 Shore D, which agrees with the value indicated by Teymoorzadeh and Rodrigue [32]. All the composite samples presented higher hardness values than the PLA. The highest hardness was obtained for PSG (84.5 Shore D, corresponding to a 10% increase in hardness compared with PLA). The PS and PG samples presented the same Shore D hardness (83). Similar results were obtained by other authors, who claimed that hardness increased when either GNP or SCF was added to polymer-based composites [33,34,35,36,37,38]. 

The composites reinforced with the Sn89-Zn8-Bi3 alloy (PSA, PSGA, PGA, and PA) showed lower hardness values (79 to 81) when compared to the other composites.

Figure 12a,b illustrates the variation of the COF as function of time (maximum of 300 s) curves of the PLA, PG, and PA samples, as well as the average coefficient of friction of all of them, respectively. These COF curves were selected because they corresponded to three different tribological behaviors.

The PLA sample showed a relatively unstable curve, with a mean COF of 0.5. The same was true for the PA sample, with a COF curve varying from 0.53 to 0.6 and a mean value of 0.55. This was the highest value among all the samples, corresponding to a 10% increase, compared to the cast PLA sample. On the contrary, the curve of the PG sample was much more stable, with the lowest mean COF of 0.43, which is 14% lower, compared to the non-reinforced PLA sample. The reason behind this is that graphene is a solid lubricant [39]. The load applied during the wear test causes the carbon flakes to separate from the composite matrix; thus, some of the surface film is transferred and acts as a solid lubricant. This results in lower COF than the PLA [40]. All the other cast samples (PGA, PSA, and PS) presented mean COF values between the values of the PG and PA samples and slightly lower than the value of the PLA. 

In line with this study, Batakliev [12] also observed a COF reduction in PLA composites reinforced with GNP and carbon nanotubes. However, the values obtained in that study (0.08 to 0.16) were much lower than the ones obtained in the present work (0.43 to 0.55). The main reason for this discrepancy in results is related to the difference in the contact pressure in the tribological tests. In Batakliev’s work [12], 6.35 mm diameter chrome SS balls and an applied load of 2 N were used, while in our study, the load applied was 5 N, and the balls had a 5 mm diameter. This meant that the contact pressure in our study was higher, leading to higher COF values [41,42]. The type of the tribological test, testing environment, material of the counterpart, and surface finish of the sample, among others, are other parameters that affect the COF. 

When a steel counterpart slides over a polymer, the polymer may soften locally or even melt, due to the temperature increase within the interface at the surface, allowing molten materials to transfer to the counterpart’s surface and form a transfer layer [43,44,45]. By switching the contact mode from metal-on-polymer to polymer-on-polymer during the sliding, the transferred polymer layer can lower the friction coefficient [46]. 

The addition of carbon-based fillers to polymers enhances the composites’ thermal stability because they have a substantially higher thermal conductivity than the polymer matrix [47]. Therefore, by dispersing the frictional heat inside the matrix, the improved thermal stability of the composites prevents the transfer of material to the counterpart surfaces. Moreover, the ability of carbon-based fillers to act as solid lubricants is well-documented [46]. During sliding contact, the surface wear of the composite releases carbon particles, which freely roll or slide under a lateral force to reduce the composite’s friction coefficient by reducing the contact between the steel ball and the composite sample’s surface [48]. The mechanical strength of the samples also affects the COF and wear [49,50]. A polymer’s inadequate mechanical strength may lead to a higher friction coefficient by facilitating contact with its counterpart, significant wear of the polymer due to micro-plastic deformation, and micro-cutting produced by the asperities on the counterpart’s surface [49,50]. Therefore, it can be concluded that the lower COF values of the carbon-reinforced PLA samples, compared to PLA, resulted not only from the lubricating effect of the fillers but also from their increased hardness. On the contrary, higher COFs were obtained for the PLA samples reinforced with Sn89-Zn8-Bi3. This can be explained by the interfacial shear strength between the two surfaces in contact [51,52]. Since polymers have lower surface energy than metals, their adhesive strength is lower than that of a metal–metal adhesive system [53]. The cohesive character of materials is a product of their atomic forces, and the adhesion of one metal to another depends on the electronic configuration of the atoms involved. Therefore, factors affecting the metal–metal adhesion depend on the atomic binding energies, distribution of the atomic species in the surface layers, surface structure, and crystal lattice orientation. Metals that lack electrons in the *s* or *p* orbitals are more likely to form adhesive connections with other metals and alloys [54]. Sn has the atomic number 50, with an electronic configuration of 5*s*^2^5*p*^2^ in the outermost shell. As a result, the addition of Sn89-Zn8-Bi3 to the PLA developed much stronger adhesion forces with the steel balls during the sliding motion, leading to higher friction. In fact, SEM analysis showed that the chipping of the PA composite occurred during the tribological tests, which may also be a result of a deficient mechanical bonding (adhesion) of the alloy to the PLA granules.

The 3D profilometry images of the wear scars and the corresponding wear profiles of the samples after the tribological tests are presented in Figure 13. The corresponding specific wear rates are illustrated in Figure 14. Figure 15 shows the SEM images of the wear scars of the samples. The PSG and PG samples presented the lowest wear depth values (~6 and 11 µm, respectively), significantly lower than the wear depth value of the PLA sample (~40 µm). The PA sample presented the highest maximum wear depth of ~110 µm.

Concerning the specific wear rates, the PLA sample presented a value of 1.8 × 10^−3^ mm^3^/N·m. The PSG sample had the lowest value (1.5 × 10^−4^ mm^3^/N·m), reduced by 12 times compared to PLA. The PG sample also showed a low specific wear rate (2.4 × 10^−4^ mm^3^/N·m). The PA sample showed the highest value (3.5 × 10^−3^ mm^3^/N·m), two times higher than the PLA. All but the PSGA samples reinforced with Sn89-Zn8-Bi3 showed lower wear resistances than PLA. This was particularly true for the PA sample with no GNP or SCF. Its low adhesion to the PLA granules may be the reason for this behavior. An increase in the wear resistance by integrating GNP into the PLA matrix was reported by Ustillos et al. [55]. In the same way, Friedrich [56] showed that SCF can effectively increase the wear resistance of PEEK + PTFE engineering polymers. Both SCF and GNP were harder than PLA, which led to a higher wear resistance. Moreover, SCF increased the higher load-bearing capacity contact area, resulting in a higher wear resistance for these composites. 

The tribological behavior of the different samples was assessed by SEM analysis (Figure 15). In accordance with the work of Lancaster [57], the predominant wear mechanism of the PLA sample was adhesion with debris release, resulting from the plastic deformation and shear. Abrasion was the predominant wear mechanism of all the reinforced PLA composites, along with adhesion. During the sliding tests, the reinforcement particles came out of the composites and rolled between the sliding surfaces, resulting in a three-body abrasive wear mechanism. Concerning the PLA composites reinforced with SCF and GNP, abrasive lines on the wear tracks were visible. In these samples, the SCF and GNP particles released from the composite during the friction test acted as solid lubricants, reducing the direct contact between the steel balls and the composites, with a consequent effect on the friction coefficients and wear. A similar phenomenon has already been reported in literature [16,47]. Moreover, no obvious interfacial failure was detected in these samples. The SCF and GNP fillers disrupted the creation of a network of microcracks by reducing the concentration of stress during the sliding tests, which drastically lowered the wear [58]. 

The PLA composites with Sn89-Zn8-Bi3 (PA, PSA, PGA, and PSGA) showed cracks, grooves, and chipping/delamination, due to the lack of adhesion between the PLA matrix and the metallic reinforcement. This led to a higher material loss, as well as a higher wear rate than all the other samples. Some cracks were detected in these composites. The repeated sliding of the counter body caused surface fatigue of the polymer through the initiation and propagation of micro-cracks into the subsurface. This process causes geometrical alterations to the surface topography [59]. 

Finally, the values of the specific wear rate and COF calculated can be correlated with hardness (Figure 16). As previously mentioned, there was a tendency for the COF and specific wear rate values to decrease as the hardness increased. The joint addition of GNP and SCF proved to be a good option to fabricate PLA-based composites with high hardness, a low COF value, and high wear resistance.

## 4. Conclusions

Mechanical alloying followed by casting proved to be a suitable method to fabricate PLA-based bio-composites reinforced with SCF, GNP, and Sn89-Zn8-Bi3. The addition of carbon-based reinforcements to PLA increased its hardness and tribological behavior (lower COF and specific wear rate). The highest Shore D hardness was obtained for the PSG sample (84.5, corresponding to a 10% increase compared to PLA). The PG sample showed the lowest COFs (0.43), while the PSG sample presented the highest hardness and the lowest specific wear rate (85 and 1.5 × 10^−4^ mm^3^/N·m, respectively). The PLA samples reinforced with Sn89-Zn8-Bi3 did not perform well compared to the other composite samples, showing even worse tribological behavior than PLA.

## Figures and Tables

**Figure 1 materials-16-01608-f001:**
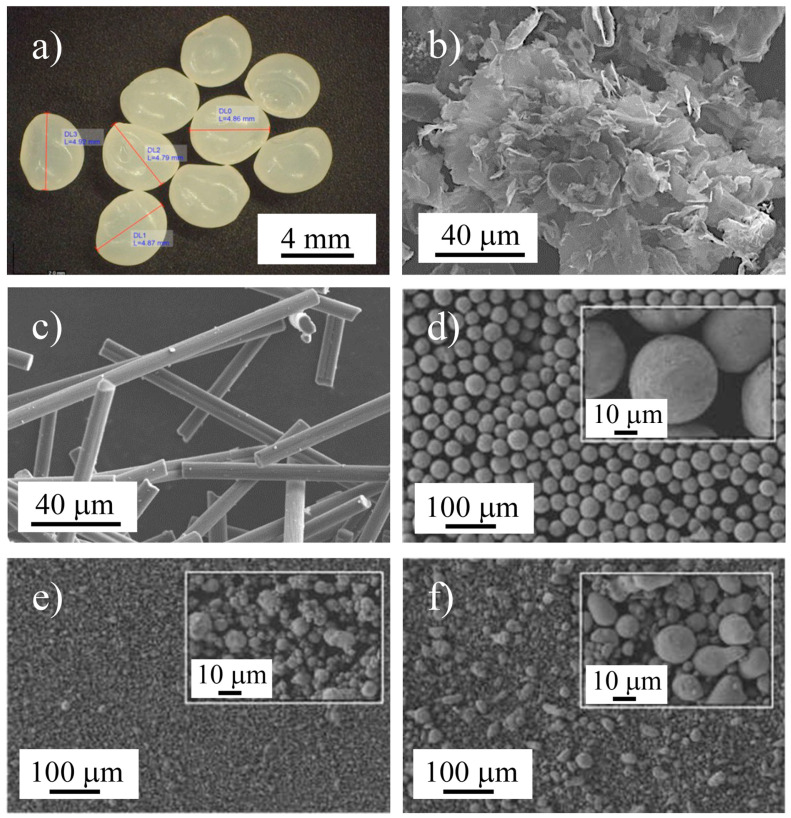
(**a**) Optical image of the PLA granules, and (**b**–**f**) SEM images of the GNP, SCF, Bi, Zn, and Sn raw materials, respectively.

**Figure 2 materials-16-01608-f002:**
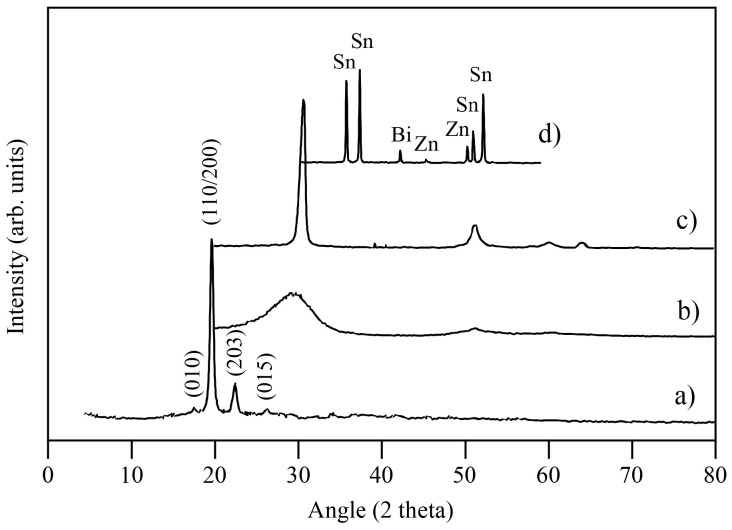
XRD patterns of the raw materials used in this study. (**a**) PLA, (**b**) SCF, (**c**) GNP, and (**d**) MA’ed Sn89-Zn8-Bi3 mixture.

**Figure 3 materials-16-01608-f003:**
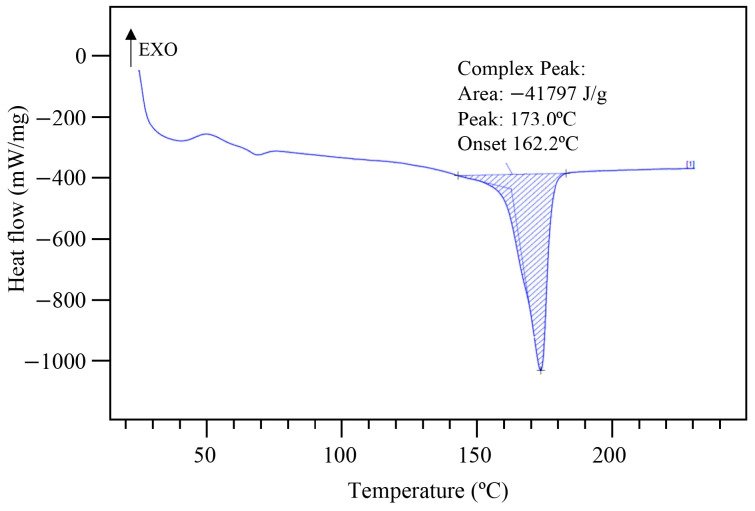
DSC curve of raw PLA.

**Figure 4 materials-16-01608-f004:**
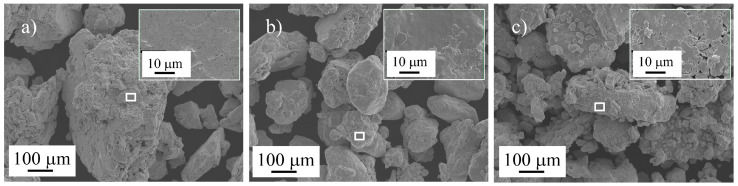
SEM images of the Sn89-Zn8-Bi3 mixture after (**a**) 5 h, (**b**) 10 h, and (**c**) 25 h of MA.

**Figure 5 materials-16-01608-f005:**
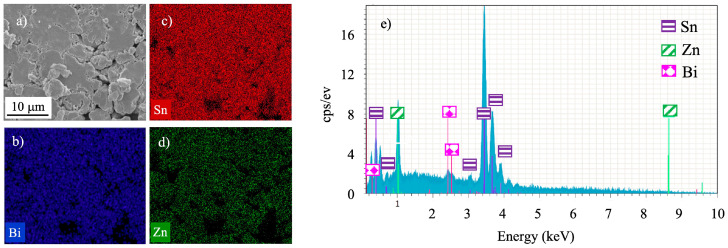
(**a**) SEM image and (**b**–**e**) EDS analysis of the Sn89-Zn8-Bi3 mixture after 25 h of MA.

**Figure 6 materials-16-01608-f006:**
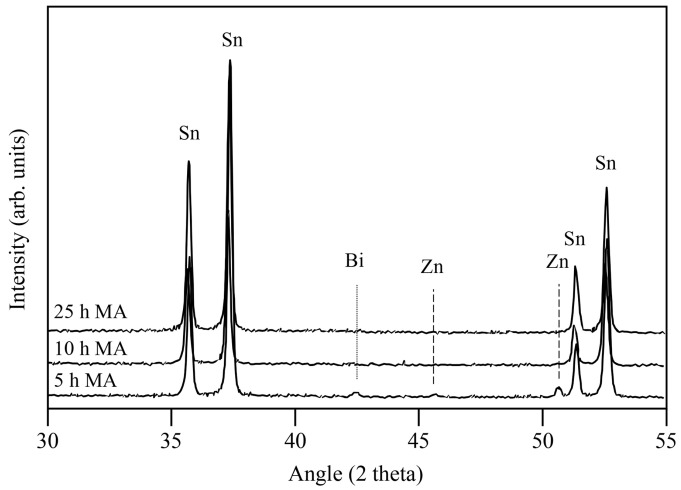
XRD patterns of the Sn89-Zn8-Bi3 mixture after 5, 10, and 25 h of MA.

**Figure 7 materials-16-01608-f007:**
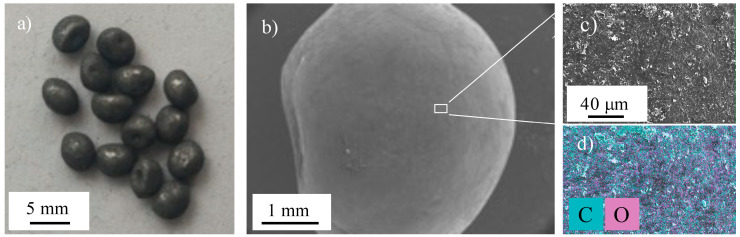
(**a**) Optical micrographs of the PLA granules coated with GNP and Sn89-Zn8-Bi3, (**b**,**c**) SEM images of the PLA granules coated with GNP and SCF, and (**d**) EDS elemental maps of the PLA coated granules shown in (**b**,**c**).

**Figure 8 materials-16-01608-f008:**
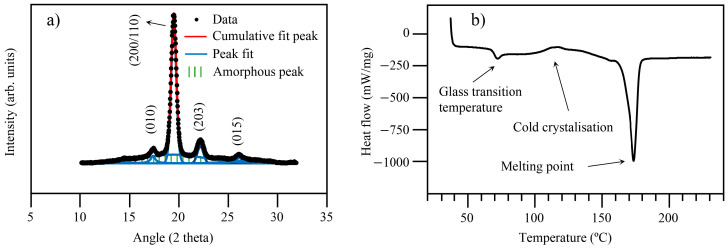
(**a**) XRD peak deconvolution and (**b**) DSC results from the cast PLA.

**Figure 9 materials-16-01608-f009:**
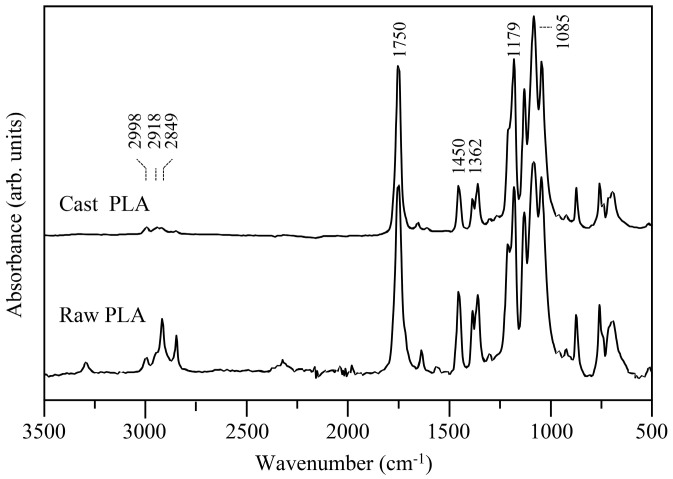
FTIR results of the raw and cast PLA.

**Figure 10 materials-16-01608-f010:**
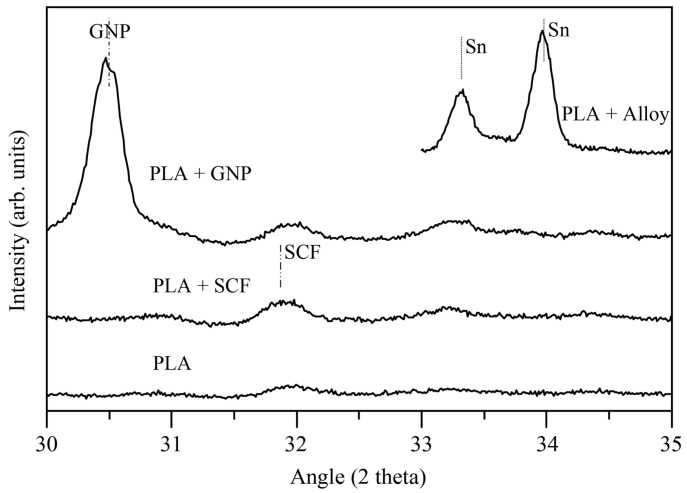
XRD patterns of the non-reinforced and reinforced cast PLA samples.

**Figure 11 materials-16-01608-f011:**
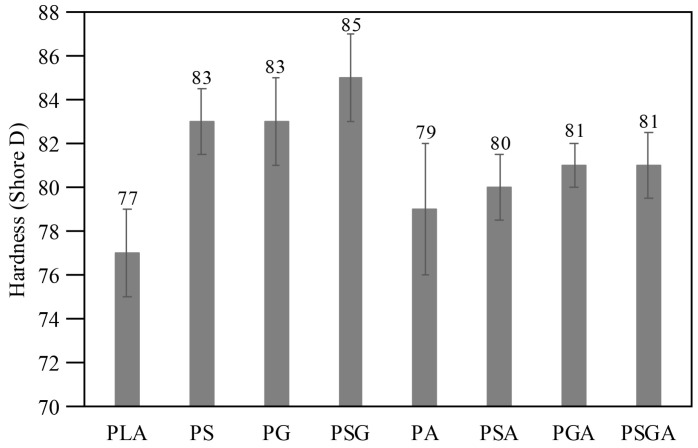
Shore D hardness of the non-reinforced and reinforced cast PLA samples.

**Figure 12 materials-16-01608-f012:**
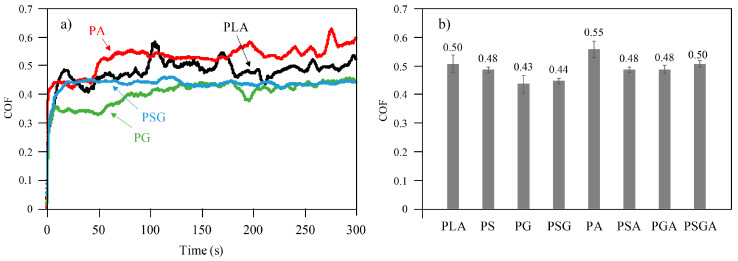
(**a**) COF vs time curves of the PLA, PG, PA, and PSG samples and (**b**) the average COF values of non-reinforced and reinforced cast PLA samples.

**Figure 13 materials-16-01608-f013:**
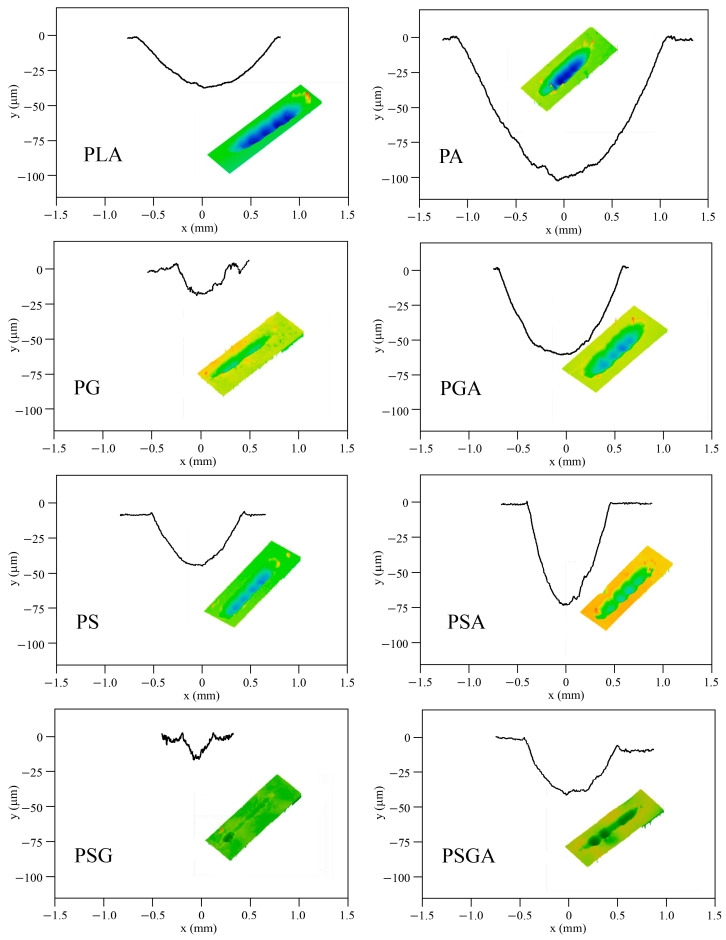
Wear profiles and 3D scans of non-reinforced and reinforced cast PLA samples after the tribological tests.

**Figure 14 materials-16-01608-f014:**
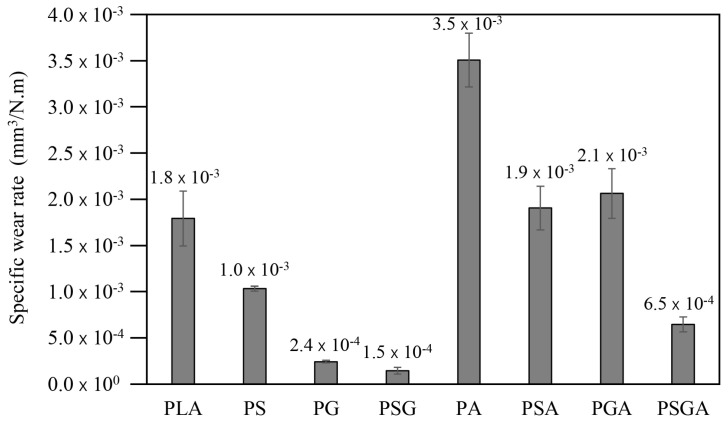
Specific wear rates of the non-reinforced and reinforced cast PLA samples.

**Figure 15 materials-16-01608-f015:**
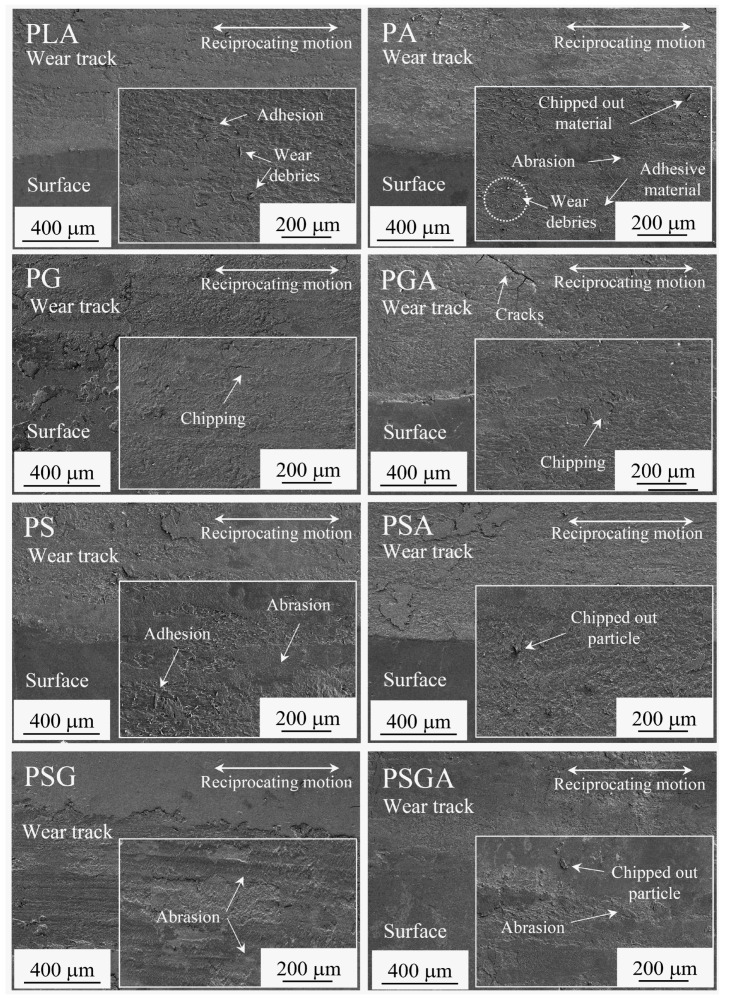
SEM images of the wear scars of the non-reinforced and reinforced cast PLA samples after the tribological tests.

**Figure 16 materials-16-01608-f016:**
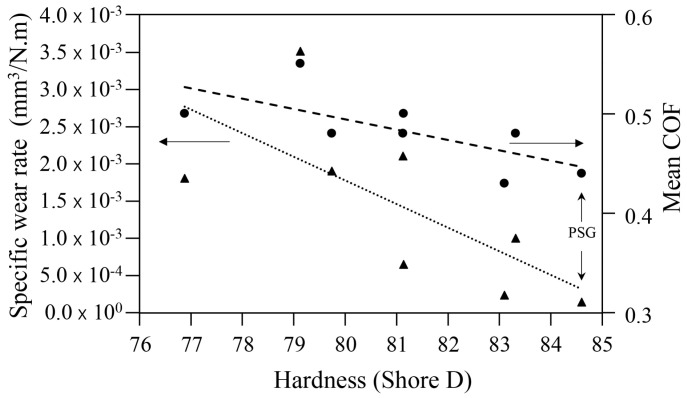
Hardness versus specific wear rate and COF of the non-reinforced and reinforced cast PLA samples.

**Table 1 materials-16-01608-t001:** Samples produced during this research.

Sample	Short Name	Description
PLA	PLA	Cast PLA.
PLA + SCF	PS	Cast from PLA granules coated with short carbon fibers by MA.
PLA + GNP	PG	Cast from PLA granules coated with graphene nanoparticles by MA.
PLA + SCF + GNP	PSG	Cast from PLA granules coated with short carbon fibers and graphene nanoparticles by MA.
PLA + Sn alloy	PA	Cast from PLA granules coated with Sn_89_-Zn_8_-Bi_3_ by MA.
PLA + SCF + Sn alloy	PSA	Cast from PLA granules coated with Sn_89_-Zn_8_-Bi_3_ + short carbon fibers by MA.
PLA + GNP + Sn alloy	PGA	Cast from PLA granules coated with Sn_89_-Zn_8_-Bi_3_ + graphene nanoparticles by MA.
PLA + SCF + GNP + Sn alloy	PSGA	Cast from PLA granules coated with Sn_89_-Zn_8_-Bi_3_ + short carbon fibers + graphene nanoparticles by MA.

## Data Availability

Corresponding author B.T.

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
