# Peer review of "A Comparative Study of Different Poly (Lactic Acid) Bio-Composites Produced by Mechanical Alloying and Casting for Tribological Applications"

_materials, 2023, doi:10.3390/ma16041608_

Round 1

Reviewer 1 Report

The work "A comparative study of different poly(lactic acid) bio-composites produced by mechanical alloying and casting for tribological applications" deals with the tribological characterization of poly(lactic acid) matrix (PLA) composites reinforced with graphene, carbon fiber and tin alloy. Although the use of carbonaceous fillers has also been widely used with PLA, this study produces some interesting results on the friction phenomena and the abrasion mechanisms of biocomposites. To improve the quality of the work, it is suggested to correct some inaccuracies present in the current version:

Line 85: change powers with powders

Lines 88-89, 102, 118, 227: the superscript and subscript numbers must be used in the dimensions (cm3, N2, CH3)

Table 1: change PLA+GNP+GNP with PLA+SCF+GNP

Line 94: The quantity of filler used with respect to the matrix (% wt.) is not reported. This information is decisive for the results obtained and in relation to the mechanical compounding method used.

Line 102: Is this the correct mold size? Seems small for the studies reported

Equation 1: the fraction lacks parentheses (so the formula is ambiguous)

Line 200: greek m in thickness dimension

Figure 11: How is the error reported? It doesn't look like a standard deviation because the bar looks skewed.

Figure 12a: the curves are difficult to distinguish; it would be useful to use dashes or different colors or insert symbols.

Figure 13: Although the reported results are evident and absolutely interesting, a greater uniformity of scale would make the comparison of the graphs easier.

A general revision of the English, making the study more scientific and less discursive, would be helpful.

With appropriate changes made in the form of a minor revision, the manuscript is recommended for publication.

Author Response

The authors would like to thank the reviewer's comments that greatly contributed to the improvement of the paper.

1. Line 85: change powers with powders

This was corrected in the new version of the text.

2. Lines 88-89, 102, 118, 227: the superscript and subscript numbers must be used in the dimensions (cm3, N2, CH3).

This was corrected in the new version of the text.

3. Line 94: The quantity of filler used with respect to the matrix (% wt.) is not reported. This information is decisive for the results obtained and in relation to the mechanical compounding method used.(

In line 196 of the manuscript it is written that the volume ratio between the PLA and the fillers of 1:0.02, which corresponds to a filler volume of 2 %.

4. Line 102: Is this the correct mold size? Seems small for the studies reported There was a mistake. The diameter of the mould was 4 cm and not 4 mm.

This was corrected in the new version of the text.

5. Equation 1: The fraction lacks parentheses (so the formula is ambiguous

This was corrected in the new version of the text.

6. Line 200: greek m in thickness dimension. 

This was corrected in the new version of the text.

7. Figure 11: How is the error reported? It doesn't look like a standard deviation because the bar looks skewed.

The error bars meant the highest and lowest values obtained. In the new figure the standard deviation is shown.

8. Figure 12a: the curves are difficult to distinguish; it would be useful to use dashes or different colors or insert symbols.

In the new figure, the curves are in colour. 

9. Figure 13: Although the reported results are evident and absolutely interesting, a greater uniformity of scale would make the comparison of the graphs easier.

The new figure presents the same scale for all the profiles.

10. A general revision of the English, making the study more scientific and less discursive, would be helpful. 

The paper was revised by an English native.

Reviewer 2 Report

Review

The work is devoted to the actual problem of obtaining new composites based on PLA. However, the application of carbon nanoparticles and metal compounds can’t be called innovative. There is a lot of research on this issue. Nevertheless, the work has some novelty. There are some issues to be addressed.

The abstract should be improved and supplemented with numerical data. Lines 9-10: Why is the information here?  It’s common knowledge.

The X-ray results could have been better described. What is the form (α, α’ etc.) of crystallites in the studied PLA? It can be discussed whether it changes in the process of modifying the PLA. Figure 8 should be made clear. Please number the diffractograms and make changes to the description of the figure.

FTIR results: Lines 222-228: why is a comma used for 2.850 cm-1? It should be 3000-2850 cm-1. The bonds relating to the crystalline and amorphous phases of PLA are not described. Some references should be added.

Figure 15: There is very small the font size on the micrographs ( especially in 200 μm) inserts. Please increase the font size.

Author Response

The authors would like to thank the reviewer's comments that greatly contributed to the improvement of the paper.

1. The abstract should be improved and supplemented with numerical data. Lines 9-10: Why is the information here?  It’s common knowledge.

The abstract was improved.

2. The X-ray results could have been better described. What is the form (α, α’ etc.) of crystallites in the studied PLA? It can be discussed whether it changes in the process of modifying the PLA. Figure 8 should be made clear. Please number the diffractograms and make changes to the description of the figure.

Figure 8a was improved and some more information was included in the text.

3. FTIR results: Lines 222-228: why is a comma used for 2.850 cm-1? It should be 3000-2850 cm-1. The bonds relating to the crystalline and amorphous phases of PLA are not described. Some references should be added.

This has been corrected in the new version of the manuscript. Additional information was included in Fig. 9.

4. Figure 15: There is very small the font size on the micrographs ( especially in 200 μm) inserts. Please increase the font size.

The font size was increased in the new figure.

Reviewer 3 Report

The present manuscript investigates the use of polylactic acid for tribological applications using mechanical alloying and casting. The results presented in the manuscript are interesting and the manuscript is well written. The manuscript can be accepted for publication after a few minor revisions. 

1. What is the meaning of amorphous peaks in the XRD spectra of Figure 8? To me, it seems quite strong. Can an amorphous structure give such a peak?

2. Why C-H stretch peak in FTIR spectra of cast PLA was weakened? 

Author Response

The authors would like to thank the reviewer's comments that greatly contributed to the improvement of the paper.

1. What is the meaning of amorphous peaks in the XRD spectra of Figure 8? To me, it seems quite strong. Can an amorphous structure give such a peak?

This amorphous phase was formed during cooling from the casting temperature. The amorphous peak can be of some intensity and it is quite large. Please ee e.g. Int. J. Mol. Sci. 2012, 13, 5878-5898; doi:10.3390/ijms13055878

2. Why C-H stretch peak in FTIR spectra of cast PLA was weakened? 

We are not sure that a weakening of the C-H stretch band occurred during casting. The intensity of the C-H bending peak around 1450 and 1360 cm-1 is also lower than raw PLA. In order to place both spectra in the same figure the normalization wrongly induces the perception of weak C-H stretching. Anyway, additional work is being done to fully understand if some happened during PLA processing.

Round 2

Reviewer 2 Report

No more critical comments.